# Combined Wind Lidar and Cloud Radar for High-Resolution Wind Profiling

José Dias Neto[1], Louise Nuijens[1], Christine Unal[1], and Steven Knoop[2]

[1]Delft University of Technology, Delft, The Netherlands
[2]Royal Netherlands Meteorological Institute, De Bilt, The Netherlands

**Correspondence:** José Dias Neto (J.DiasNeto@tudelft.nl)

**Abstract.** This paper introduces an experimental setup for retrieving horizontal wind speed and direction profiles with a high temporal and vertical resolution for process studies and validation of convection-permitting model simulations. The CM-TRACE (Tracing convective momentum transport in complex cloudy atmospheres) campaign used collocated wind lidar and cloud radar measurements to retrieve seamless wind profiles from near the surface up to cloud tops. It took place in Cabauw, the Netherlands, between September 13[th] and October 3[rd] 2021. The intermediate processing steps for generating the Level 1 and Level 2 data, such as second trip echos filtering, offset correction, wind retrieval, re-gridding and flagging, are described. In Level 1 (https://doi.org/10.5281/zenodo.6926483, Dias Neto (2022a)), the data from lidar and radars are kept in the original spatial and temporal resolution, while in Level 2 (https://doi.org/10.5281/zenodo.6926605, Dias Neto (2022b)), they are regridded to a common spatial and temporal resolution. Statistical analyses of the lidar's and radar's wind speed and direction profiles indicate a correlation higher than 0.95 for both variables. The bias of wind direction and speed calculated between radar's and lidar's observations are 0.24° and -0.16 $\mathrm{ms^{-1}}$, respectively. The foreseen initial application of the datasets includes the study of convective momentum transport and its validation in regional weather forecasts and large-eddy simulation hindcasts.

## 1 Introduction

Wind is an essential component in nearly every weather phenomenon on Earth through its transport of heat, moisture and scalars. How winds blow sets patterns of precipitation on large scales through *e.g.* atmospheric rivers or monsoons (Gimeno et al., 2014; Zemp et al., 2014; Naakka et al., 2019; Gimeno et al., 2020), while on small scales it influences surface heat and moisture fluxes, convection and cloud development. Large-scale wind is driven by thermal (pressure) gradients, but modified by a range of small scale processes, including surface drag, momentum transport and gravity waves. The parameterization of those small-scale processes in weather and climate models remains uncertain and persistent wind biases likely related to such processes continue to exist.

Wind observations for data assimilation and model validation are therefore invaluable, but generally limited to the surface layer where meteorological stations (over land) or permanent moorings or buoys (in the ocean) exist. Besides meteorological towers (limited to 200 m) or airborne wind measurements (limited in time), ground-based radar and lidars can be used more routinely to measure wind profiles beyond the surface layer.

The so-called velocity-azimuth-display (VAD) radar approach for retrieving the wind properties was proposed by Lhermitte (1962); Browning and Wexler (1968); Lhermitte (1969), where the mean Doppler velocity (MDV) as a function of azimuth results in a sine curve. Later, Wilson (1970); Kropfli (1986) used radars to study turbulence in the boundary layer based on the VAD. Weather Doppler radars have also been extensively used by weather services (Chandrasekar et al., 2018; Kumjian, 2018) to monitor winds, mitigate the impact of storms, study the evolution of meteorological systems (e.g. tornadoes, cyclones) (Kosiba et al., 2013) and detect wave structures in the horizontal wind (Miller et al., 2022).

In the last decade, due to the current global transition from fossil fuel power plants to systems based on renewable energy sources, the wind energy industry has bloomed. For accurate wind power prediction, understanding the influence of turbulence in the atmospheric boundary layer, as well as topography and surface drag, is crucial. Wind turbulence can strongly affect energy production (Elliott and Cadogan, 1990; Peinke et al., 2004; Clifton and Wagner, 2014) or even damage the wind turbines (Kelley et al., 2006). Studying such processes requires much higher resolution wind observations to understand the temporal and spatial scales with which wind varies, including over ocean. This has spurred the development and deployment of commercial Doppler wind lidar. Based on VAD radar works, Eberhard et al. (1989) used lidar for studying turbulence for the first time, and since then, lidar has been largely used for this application (Newman et al., 2016; Newman and Clifton, 2017; Mann et al., 2010; Sathe et al., 2015; Smalikho and Banakh, 2017; Bonin et al., 2017). Sathe and Mann (2013) provide a good review of lidar-based experiments.

Recently, observations from wind lidar and weather Doppler radar at low elevations (2 °) were combined to extend the range of retrieved horizontal wind (Ritvanen et al., 2022). But few studies combine radar and lidar to investigate the detailed evolution of wind below and within clouds. One exception is the work from Bühl et al. (2015), where the authors combine radar and lidar observations, but only for retrieving the vertical component of the wind.

In this study, we attempt to develop a dataset that uses clouds explicitly to make extended wind profiles throughout the lower atmosphere, and not just in the surface layer. Clouds have long helped visualise and quantify the winds at higher levels (*e.g.* atmospheric motion vectors can be derived from clouds' motion Velden et al. (2005); Velden and Bedka (2009); Kishtawal et al. (2009); Cordoba et al. (2017)). But how winds themselves are modified by convection remains poorly studied, let alone observed. By measuring winds below and through clouds, we may gain insight into one of the main uncertainties for wind prediction as highlighted by the numerical weather prediction community: namely, convective momentum transport (CMT), which we broadly define as convectively-driven transport of momentum through updrafts, downdrafts and the cloud-scale or even mesoscale circulations that accompany clouds.

Turbulent eddy-resolving models may lend themselves better than point-measurements for the study of momentum transport by providing a three-dimensional view of the multi-scale flow. However, they are limited by the model's periodic boundary conditions, the use of domains smaller than the scales of mesoscale cloud organisation observed in nature, and possible mis-representations of turbulence and convection. Recent Large-Eddy Simulations (LES) studies with open boundaries and large domains show that horizontal flows can generate substantial momentum fluxes (Dixit et al., 2021), which are not present in more traditional (BOMEX, Siebesma and Cuijpers (1995); RICO, vanZanten et al. (2011)) cases of shallow cumulus convection. They also show that such flows produce so-called counter-gradient transport, whereby the environmental wind shear is

enhanced instead of diminished (due to local downgradient turbulent diffusion). The handful of LES studies focusing on CMT have been carried out for convection over oceans, while studies of CMT over land are limited.

Recently, Koning et al. (2021) combined 9 years of wind observations from a 200 m tall tower and LES from a commercialised graphics processing unit (GPU) version of Dutch Atmospheric Large-Eddy Simulation (DALES, Heus et al. (2010)) over Cabauw (The Netherlands) to investigate the relationship between different cloud regimes (clear sky, shallow clouds, non-convective clouds) and momentum flux and wind shear in the boundary layer. The authors found that clear sky and shallow clouds days have a similar diurnal cycle of near-surface winds, but the further deepening of the convective boundary layer in the presence of clouds can lead to a larger daytime increase in near-surface winds. Furthermore, for a similar atmospheric stability in the surface layer, days with shallow clouds sustained larger surface momentum fluxes for a given wind gradient. The data also suggests that more crosswind momentum fluxes are present within the mixed layer - hinting at more organized cloud or roll cloud structures - compared to days when non-convective clouds are predominant. As part of the Ruisdael Observatory (https://ruisdael-observatory.nl), the Dutch LES will be run over the entire Netherlands at a 100 m grid daily. To accompany and validate these simulations high resolution (wind) measurements are invaluable.

This paper introduces an experimental setup where scanning cloud radar and wind lidar observations are combined to retrieve horizontal and vertical wind profiles with the highest possible resolution. Even though the sonic detection and ranging (SODAR) and radar wind profiler (RWP) instruments can have similar vertical resolutions as cloud radar and lidar, the SODAR and RWP averaging time is around 10 min, which limits the study of turbulence and convection. By merging radar and lidar observations, this experiment provides, for the first time, continuous profiles of the horizontal wind from near the surface up to cloud tops. The experiment is conducted as part of the Dutch Research Council (NWO) funded Tracing Convective Momentum Transport in Complex Cloudy Atmospheres experiment project (CMTRACE) that targets convective momentum transport under different cloud conditions and across different temporal/spatial scales.

## 2 Experiment

CMTRACE occurred between September 13[th] and October 3[rd] 2021, in Cabauw, The Netherlands. It was carried out by the Delft University of Technology and the Royal Netherlands Meteorological Institute, in coordination with the Ruisdael Land-Atmosphere Interactions Intensive Trace-gas and Aerosol measurement campaign (RITA, https://ruisdael-observatory.nl/the-rita-2021-campaign), which took place at the same site, allowing the synergy of combining observations from in situ measurements (ground-based and airborne) with the CMTRACE remote sensing observations. For this experiment, one wind lidar and two cloud radars were installed close to each other; the wind lidar provided information on the sub cloud layer winds, while the cloud radars provided information on the cloud layer winds. The horizontal distance between them was less than 60 m (Figure 1). This small separation between the instruments was intended to optimize overlap in sampled volume.

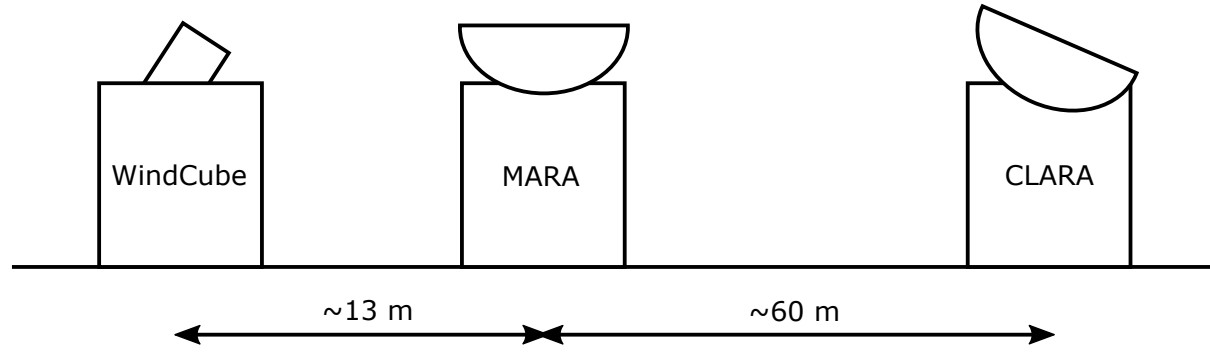

**Figure 1.** Conceptual illustration (not to scale) of the horizontal separation between the lidar and radars operated during CMTRACE.

### 2.1 Instruments

The first cloud radar is a dual-frequency (35 and 94 GHz) scanning polarimetric frequency-modulated continuous-wave radar (FMCW) produced by Radiometer Physics GmbH (hereafter CLARA, CLoud Atmospheric RAdar). This radar allows setting different configuration parameters for different range intervals (*e.g.* range resolution, integration time). During the campaign, three range intervals were used, and the radar settings from each range are summarized on Table 1 and Table 3. CLARA was operated performing clockwise and anti-clockwise periodical Plan Position Indicator scans (azimuths: 0-359.9°) with an elevation angle of 75°; each scan sequence lasted for about 72 s. This high elevation was chosen to minimize the MDV folding effect that could affect the observations if the scatterers' velocities were larger than the Nyquist velocity. The Nyquist velocity for each range interval is listed in Table 3. Although CLARA is a dual-frequency system, only the Doppler velocities from 35 GHz are used due to an approximately three times larger Nyquist velocity when compared to the 94 GHz.

The second cloud radar is a single frequency (94 GHz) scanning polarimetric frequency-modulated continuous-wave radar (FMCW) produced by Radiometer Physics GmbH (hereafter MARA, Mobile Atmospheric RAdar). Although MARA is a scanning capable radar, it continuously pointed vertically during the experiment, providing vertical profiles with a temporal resolution of 1s. As CLARA, MARA also allows setting different range resolutions for different range intervals, and it was also operated using three range intervals. The configuring parameters for each range interval are also listed in Table 1 and Table 2.

The wind lidar is a WindCube-200s scanning lidar produced by Vaisala (hereafter WindCube). It is a pulsed system operating at a wavelength of 1.54 μm, and it is capable of scanning at different azimuths and elevations. This lidar also allows defining different integration times (from 0.1 up to 10 s) and range resolution (from 25 up to 100 m). During the campaign, the WindCube was operated following the six-beam scanning strategy proposed by Sathe et al. (2015), which, according to the authors, provides more information about turbulence than the VAD. The six-beam scanning strategy consists in measuring the radial velocity at five azimuths equally separated by 72° at a specific elevation angle and one additional measurement at 90° elevation (see Sathe et al. (2015) for a complete description). During CMTRACE, the elevation of the slanted measurements was set to 75°, producing a conical sampling volume equivalent to that from CLARA. Table 1 lists additional configuring parameters used by the WindCube.

During the campaign, the three instruments continuously operated following the abovementioned strategy. Apart from that,
no other scanning strategy was used.

**Table 1.** Technical specifications and settings of the lidar and cloud radars operated during CMTRACE.

| Specifications | MARA[a] | CLARA[a] | WindCube |
|---|---|---|---|
| Frequency [GHz] | 94 | 35 | $1.94 \ 10^5$ |
| Chirp repetition frequency [kHz] | 14.0 - 5.8 | 12.8 - 6.4 | 20[b] |
| Doppler velocity resolution [ms$^{-1}$] | 0.03 - 0.04 | 0.05 - 0.1 | - |
| 3dB Beam Width [°] | 0.55 | 0.84 | - |
| Nyquist velocity [±ms$^{-1}$] | 11.2 - 4.6 | 27.5 - 13.7 | 41 |
| Range resolution [m] | 22.3 - 37.6 | 22.3 - 37.6 | 50 |
| Temporal sampling [s] | 1 | 1 | 2 |
| Time of a complete scan [s] | - | 72 | 22 |
| Minimum / Maximum range [m] | 111 / 11974 | 111 / 11974 | 100 / 7500 |

[a] Chirp repetition frequency, Doppler velocity resolution, Nyquist velocity and range resolution depend on
the chirp definition; those values are indicated in Table 2 for MARA and in Table 3 for CLARA.

[b] For the WindCube, this value refers to the pulse repetition frequency.

**Table 2.** Configuration parameters used by the single frequency radar MARA for each chirp sequence.

| | Chirp sequence | | |
|---|---|---|---|
| Attributes | 1 | 2 | 3 |
| Integration Time [s] | 0.034 | 0.137 | 0.274 |
| Range Interval [m] | 111.7 - 581.3 | 620.9 - 1997.9 | 2033.4 - 11974.8 |
| Range Resolution [m] | 22.3 | 26.9 | 37.6 |
| Nyquist Velocity [±ms$^{-1}$] | 11.2 | 9.9 | 4.6 |
| Doppler velocity resolution [ms$^{-1}$] | 0.04 | 0.03 | 0.03 |
| Chirp Repetition Frequency [kHz] | 14.0 | 12.4 | 5.8 |

## 2.2 Weather characteristics

The 21 days of measurements were characterized by different weather conditions and a diverse cloud cover. For a rapid
overview of the weather conditions and cloud coverage, Table 4 provides the daily estimated duration in hours of low-level
clouds (LLC), mid-level clouds (MDC), high-level clouds (HLC), deep convective clouds (DCC) and stratiform clouds (SC)
(cloud levels are described in Lamb and Verlinde (2011), Ch 1). The duration of the cloud cover was estimated by visual
inspection of data recorded by MARA. Table 4 also provides the maximum and mean precipitation rate (RR) measured by an

**Table 3.** Configuration parameters used by the dual-band radar CLARA for each chirp sequence.

| | Chirp sequence | | |
|---|---|---|---|
| Attributes | 1 | 2 | 3 |
| Integration Time [s] | 0.034 | 0.137 | 0.274 |
| Range Interval [m] | 111.7 - 581.3 | 620.9 - 1997.9 | 2033.4 - 11974.8 |
| Range Resolution [m] | 22.3 | 26.9 | 37.6 |
| Nyquist Velocity [$\pm\mathrm{ms}^{-1}$] | 27.5 | 19.2 | 13.7 |
| Doppler velocity resolution [$\mathrm{ms}^{-1}$] | 0.1 | 0.07 | 0.05 |
| Chirp Repetition Frequency [kHz] | 12.8 | 8.9 | 6.4 |

optical disdrometer and the daily mean wind speed (WS200) and direction (WD200) derived from the WindCube observations at 200 m above the surface.

Table 4 indicates that most of the days were non-precipitating with few cloud cover (predominantly LLC). During those days, the mean wind speed at 200 m above the ground ranged between 3.6 and 9.6 $\mathrm{ms}^{-1}$; the wind direction was predominantly from the southwest, but for some days, the wind direction changed to the east. During the precipitating days, DCC and SC clouds were present in addition to LLC, MLC and HLC. The wind speed for these days ranged between 6.7 and 12.9 $\mathrm{ms}^{-1}$, and the wind direction was predominately from the southwest with the exception of days where it changed to the east.

## 3 Data Processing

The CMTRACE dataset is structured in 3 levels according to the processing steps applied. Those different processing levels are designed to facilitate the usage of the CMTRACE dataset, and with that, users will have the possibility to choose the data level that better suits their needs. The processing steps for each level are summarised in Figure 2, and they are described in the following sections.

### 3.1 Processing levels

The original data output from the WindCube, MARA and CLARA is defined as the CMTRACE Level 0 dataset. In Level 0, the variables available in the dataset are related to the scatterer properties (*e.g.* backscattered signal, MDV and spectrum width), and the data from the WindCube is still affected by noise. Note that neither the lidar nor the radars datasets provide the horizontal wind speed and direction observations at this level.

The Level 1 processing starts with the removal of artefacts present in the Level 0 dataset, as indicated in the flowchart (Figure 2). A filter based on the WindCube's status variable similar to that described in the manual is applied to the WindCube observations to filter out the noise data and non-realistic MDV values. After this filtering step, the WindCube data is still affected by second trip echoes (STE) produced by clouds from altitudes further than the maximum range sampled by the WindCube. A similar issue was also found in previous experiments (Bonin and Alan Brewer, 2017; Bonin et al., 2017).

**Table 4.** Daily characterization of the cloud coverage and precipitation during CMTRACE. The first five columns (LLC, MLC, HLC, DCC, SC) indicate the approximated duration in hours of each class of clouds. RR indicates the maximum and mean precipitation rate in mm/h measured by a nearby optical disdrometer. WS200 and WD200 are the mean wind speed and direction derived from the WindCube observation at 200 m above the ground.

| date | LLC | MLC | HLC | DCC | SC | RR [mm/h] | WS200 | WD200 |
|------|-----|-----|-----|-----|-----|-----------|-------|-------|
| yyyy.mm.dd | [h] | [h] | [h] | [h] | [h] | max/mean | [m/s] | [°] |
| 2021.09.13 | 5 | 15 | 5 | 0 | 0 | 0.0/0.0 | 5.38 | 86.5 |
| 2021.09.14 | 2 | 5 | 3 | 2 | 0 | 7.6/1.38 | 7.38 | 132.35 |
| 2021.09.15 | 16 | 4 | 0 | 0 | 0 | 1.4/0.65 | 6.77 | 285.65 |
| 2021.09.16 | 1 | 0 | 1 | 0 | 0 | 0.0/0.0 | 7.26 | 295.25 |
| 2021.09.17 | 7 | 0 | 10 | 0 | 0 | 0.0/0.0 | 4.56 | 238.1 |
| 2021.09.18 | 0 | 0 | 12 | 0 | 0 | 0.0/0.0 | 5.84 | 110.38 |
| 2021.09.19 | 3 | 0 | 3 | 0 | 0 | 0.0/0.0 | 8.09 | 109.81 |
| 2021.09.20 | 7 | 0 | 6 | 0 | 0 | 0.0/0.0 | 7.57 | 81.7 |
| 2021.09.21 | 0 | 0 | 3 | 0 | 0 | 0.0/0.0 | 3.66 | 240.9 |
| 2021.09.22 | 0 | 0 | 3 | 0 | 0 | 0.0/0.0 | 6.39 | 233.46 |
| 2021.09.23 | 10 | 0 | 2 | 0 | 0 | 0.9/0.55 | 10.22 | 266.07 |
| 2021.09.24 | 11 | 0 | 0 | 0 | 0 | 0.0/0.0 | 8.5 | 239.73 |
| 2021.09.25 | 11 | 2 | 1 | 0 | 0 | 0.0/0.0 | 5.9 | 182.96 |
| 2021.09.26 | 0 | 5 | 3 | 4 | 0 | 6.3/2.01 | 6.5 | 199.42 |
| 2021.09.27 | 0 | 5 | 0 | 4.5 | 0 | 2.6/0.56 | 10.63 | 218.92 |
| 2021.09.28 | 0 | 8 | 0 | 0 | 0 | 0.0/0.0 | 9.63 | 200.22 |
| 2021.09.29 | 0 | 0 | 3 | 3 | 5 | 47.7/2.55 | 11.81 | 243.64 |
| 2021.09.30 | 6 | 0 | 11 | 1 | 1.5 | - | 11.87 | 225.6 |
| 2021.10.01 | 4 | 7 | 4 | 2 | 3 | 8.3/1.71 | 12.94 | 199.79 |
| 2021.10.02 | 1 | 7 | 0 | 0 | 12 | 11.6/1.89 | 11.38 | 188.37 |
| 2021.10.03 | 0 | 12 | 0 | 0 | 12 | 6.8/1.32 | 8.09 | 247.44 |

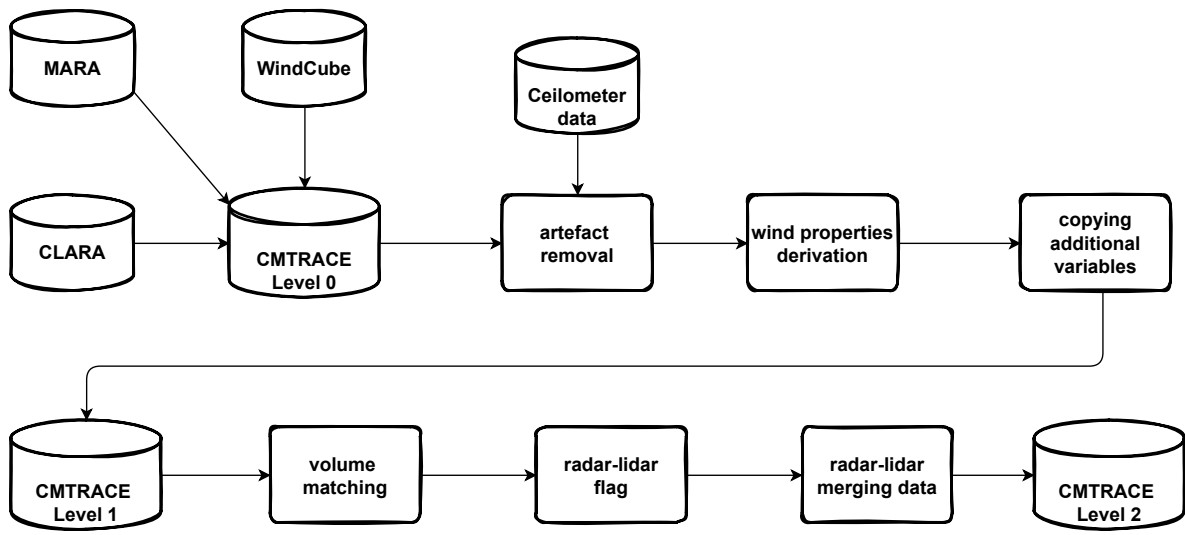

**Figure 2.** Flowchart illustrating the CMTRACE data processing: The upper part shows the processing steps applied for generating the Level 1 data, and the lower part shows the steps applied for generating the Level 2 data.

Panels a and b from Figure 4 show an example of the MDV recorded by the WindCube at $0°$ azimuth and the equivalent radar reflectivity (Ze) recorded by MARA on September $21^{st}$ 2021. The MDV values from below 2 km are continuous and distributed between -2 and 1 ms$^{-1}$ (Figure 4-a). In contrast, the MDV rapidly changes from -1 to -7 ms$^{-1}$ between 11 and 12 UTC at around 2 km. The region with fast velocities extends from 2 up to 6 km in range and appears at different times along the day. In contrast, for the same period, MARA's Ze does not show any cloud below 6 km, and the only clouds with similar shapes appear at altitudes above 8 km. A filter is then applied to minimise the presence of STE in the dataset (see subsection 3.2). After filtering, the WindCube vertical Doppler velocities are stored as the vertical wind component, and a wind retrieval based on a Fourier transform is applied to the slanted azimuthal MDV observations to retrieve the horizontal wind speed and direction (see subsection 3.3). In addition to the information related to wind, the backscattered data is also included in WindCube Level 1 data.

The Level 1 processing applied to CLARA dataset slightly differs from that applied to the WindCube observations; the radar software internally removes the noise, and for this reason, the artefact removal is skipped. The Fourier transform wind retrieval is also applied to CLARA slanted azimuthal MDV observations to retrieve the speed and direction of the horizontal wind. Surprisingly, after the retrieval, CLARA's wind profiles also had information from the lowest 2 km, where clouds are not present most of the time (see subsection 3.4). It was also noticed that an alternating offset affected the wind direction. The magnitude of this offset was estimated using the WindCube wind direction profiles as reference. It was found that each one of the range intervals listed in Table 3 was affected by a different offset (see subsection 3.5). After the offset correction, CLARA's wind speed and direction are stored as the Level 1 products. Yet, in the Level 1 processing of CLARA's data, an

index is generated to quantify the percentage of invalid data for each complete scan for each height. The invalid index value $I_{inv}$ is calculated as follows:

$$I_{inv} = 100 \frac{N_{inv}}{N_{azm}} \tag{1}$$

Where $N_{inv}$ is the number of invalid data, and $N_{azm}$ is the number of azimuths. At the current stage, it is not possible to decouple the vertical wind speed component from the hydrometeors' fall velocities using MARA's observations. However, MARA's vertical MDV is also included in Level 1. MARA's attenuated Ze is also included in Level 1, which can be used for cloud identification.

At the end of Level 1 processing, the data still have their original temporal and spatial resolution, and the data derived from
the different instruments are stored in different datasets. The Level 2 processing merges the wind profiles retrieved from the WindCube and CLARA observations and produces continuous profiles of wind speed and direction from the surface up to the boundary layer or cloud top. The vertical and temporal resolution of Level 2 data is set to 50 m (WindCube range resolution) and to 72 s (duration of CLARA PPI scans), which are the coarser resolutions in the Level 1 data. Table 5 summarizes the settings of the Level 2 dataset. The radar variables from CLARA and MARA are then interpolated to the new spatial resolution. The
temporal resolution from the WindCube and MARA data is adjusted to CLARA's temporal resolution by averaging all profiles within the time interval of each complete PPI scan from CLARA. This last strategy is used to create profiles that represent equivalent air mass volumes. After that, the processing continues by creating a time-height flag to identify the regions where the lidar and radars provide measurements. Finally, the wind speed and direction profiles from WindCube and CLARA are merged following a hierarchical criterium. The WindCube data has priority over the CLARA data, meaning that CLARA only
provides information in regions where the WindCube data is absent. However, not all data from CLARA are incorporated into Level 2. The distribution of velocity differences as a function of the of $I_{inv}$ indicates that a systematic bias becomes apparent for $I_{inv}$ larger than 50% (Figure 3). Therefore, all CLARA's data flagged with $I_{inv}$ equal to 50% or larger are not used in the Level 2 generation. In addition to the wind-related variable, Level 2 also contains the lidar and radar backscattered signals and the vertical Doppler velocities.

The complete list of variables available in CMTRACE Level 1 and Level 2 data is given in the Appendix A.

**Table 5.** General specifications of the Level 2 dataset.

| Level 2 data settings | |
| --- | --- |
| min / max range [m] | 100 / 11600 |
| range resolution [m] | 50 |
| temporal resolution [s] | 72 |

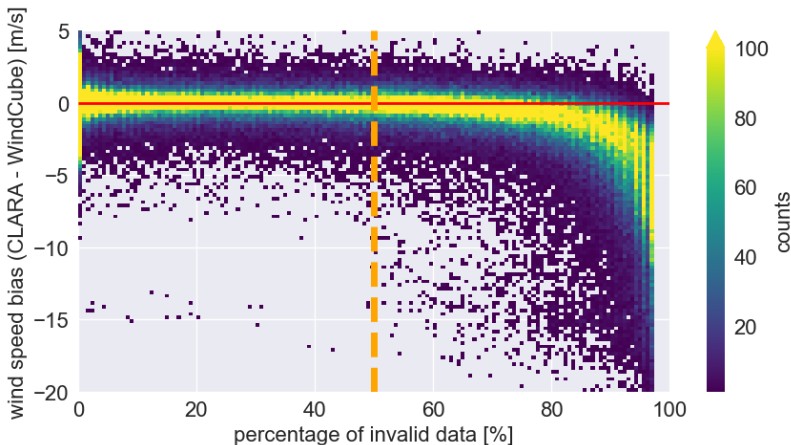

**Figure 3.** Two-dimensional histogram of the differences between the wind speed retrieved from CLARA's and WindCube's observations as a function of the invalid index; the horizontal red line indicates $0\ \mathrm{ms}^{-1}$ and the vertical dashed orange line indicates the invalid index value of 50%.

## 3.2 Second trip echoes filter

In subsection 3.1, it was introduced that the WindCube Doppler velocities observations were affected by the presence of STE produced by clouds above the maximum unambiguous range. As indicated in Table 1, the vertical range resolution of the WindCube was set to 50 m, and for the model we used, the pulse repetition frequency is automatically set to 20 kHz, limiting the maximum unambiguous range to 7.5 km. Figure 4-(a) illustrates that the STE can happen at any height. Sometimes, the STE appear in regions close to the surface (hereafter Low-STE), contrasting with the surrounding signal produced by aerosols (*e.g.* between 11 and 12 UTC below 2 km). However, at other times, the STE appear above the region loaded with aerosols (hereafter High-STE) where there is no clear contrast with the surroundings (*e.g.* signals above 2 km). In order to minimise the occurrence of Low- and High-STE, two filtering approaches were developed.

The Low-STE filter takes advantage of the contrasting characteristics. This filter is based on the temporal anomaly ($v'_{\mathrm{azm}}$) of the MDV from each azimuth angle as indicated by Equation 2; $v_{\mathrm{azm}}$ is the observed MDV, and $\bar{v}_{\mathrm{azm}}|_{\Delta t}$ is the mean value calculated within a given time window.

$$v'_{\mathrm{azm}} = v_{\mathrm{azm}} - \bar{v}_{\mathrm{azm}}|_{\Delta t} \tag{2}$$

The exact size of the time window is arbitrary. However, it should be such that a normal distribution can approximate the distribution of the anomalies (*e.g.* Figure 5); if this condition is fulfilled, the standard deviation (STD) can be used to characterise the anomaly distribution. The resulting distribution is expected to have the following characteristics: the Low-STE anomalies would populate the edges of the distribution, while the anomaly of the true signal due to turbulence would be close to the centre of the distribution. A too large or too small time window could produce an anomaly distribution that deviates from

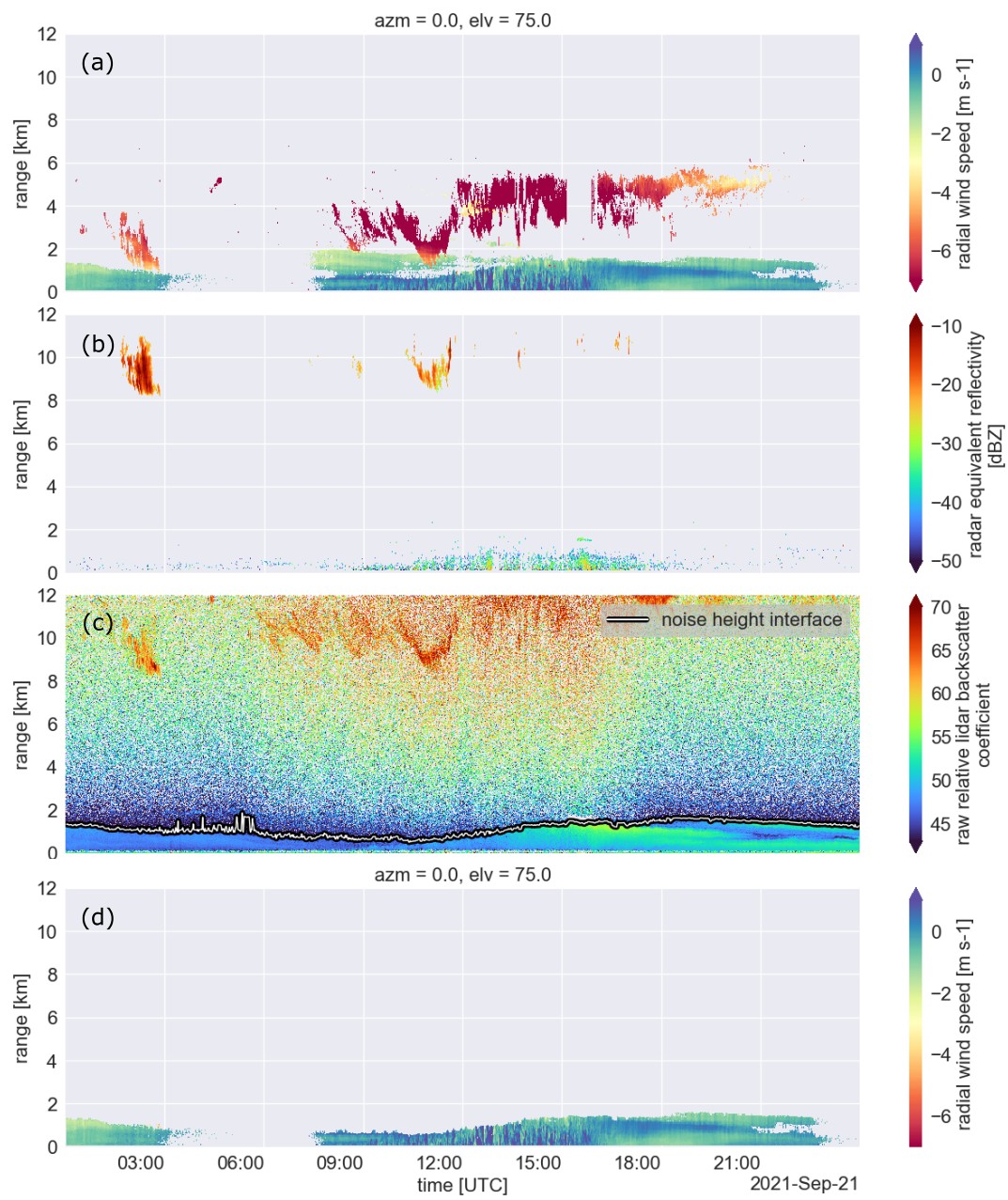

**Figure 4.** Time-height plots: (a) mean Doppler velocity measured by the WindCube, (b) radar equivalent reflectivity measured by MARA, (c) backscattered signal measured by a nearby Ceilometer, note that the negative values are already set as NaN, (d) the same as in (a), but after filtering for second trip echoes. The white shaded curve on (c) indicates the retrieved noise height interface.

a normal distribution or a distribution where the anomaly of the Low-STE populates the centre while the true signal anomaly
is near the edges.

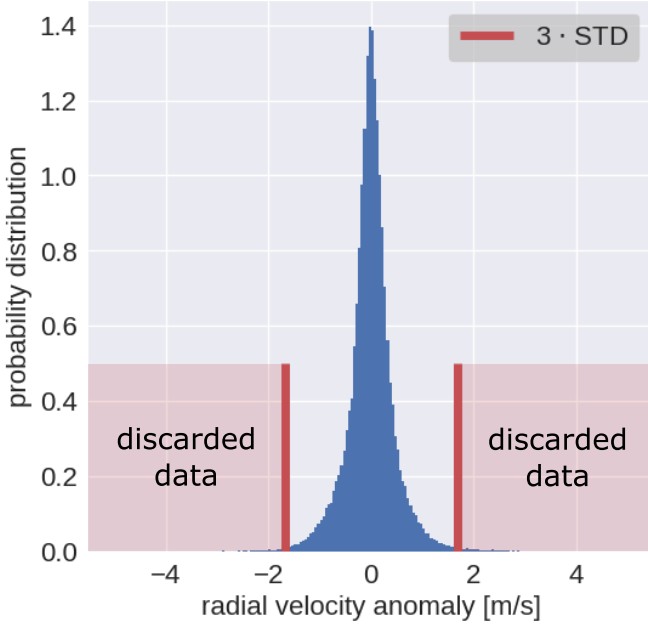

**Figure 5.** Example of the probability distribution of the MDV anomalies calculated using observations at azimuth equal to $0°$ (the same data from Figure 4-a). The vertical red lines indicate the size of the window used for filtering the data, and the light red areas indicate the values removed.

Once the anomaly STD is calculated, a window $n$ times the STD can be used to discard the data outside the defined window, as illustrated in Figure 5. The exact value of $n$ is arbitrary, but it should be a value that removes most of the STE and preserves most of the valid data. In this work, the values used for $\Delta T$ and $n$ are 3.5 hours and 3, respectively. Those values were found after several trials. Since turbulence during nighttime and daytime is different, only anomalies from between 09:00 and 16:00
UTC were used to calculate the STD. Otherwise, if the nighttime anomalies were included, the STD would be smaller than that from 09:00-16:00 UTC, and then 3·STD would still remove data not affected by STE. Note that this approach may fail in situations when the Low-STE values are comparable to the real data.

The High-STE filtering uses the backscattered signal observations from a nearby ceilometer ($\approx$20 m away from the Wind-Cube) to estimate the height interface that separates the regions loaded with lidar scatterers from clean regions. The ceilometer
observations are still affected by noise, and therefore the data from the scatterers free region is dominated by noise (Figure 4-(c)). Basically, the noise height interface (NHI) separates the non-noise region (loaded with scatterers) from the noise dominated region (scatterers free).

Because the ceilometer data did not contain the signal to noise ratio information; an alternative approach was applied to filter out the noise. It was noticed that the noise often reached negative values. This characteristic was then explored to remove the

noise from the data. The removal starts by setting all negative values as NaN (not a number). Using a moving window over the time and range coordinates, the NaNs are propagated through the noise region. For the time coordinate, the size of the window was set to cover 15 consecutive profiles, and for the range coordinate, the size of the window was set to cover 10 consecutive ranges. The NaN values were propagated by calculating the mean value of the data points covered by the moving window. The NHI is then retrieved as the largest altitude of the heights from the noise-free data in the region between the surface and 4 km. Figure 4-(c)) shows that the NHI curve closely follows the region loaded with lidar scatterers. Note that this approach will not work in regions where the noise does not reach negative values.

The detected NHI is then used as a threshold to separate the WindCube data into two regions. One region is below the NHI, where the WindCube data is expected to be predominately originated from lidar-scatterers, and the other is from above the NHI, where clouds and High-STE mainly produced the data. An example of the STE filtered data is shown in Figure 4-(d). One can see that most of the STE visible in Figure 4-(a) are no longer present in Figure 4-(d). Note that due to differences in sensitivity between the WindCube and the Ceilometer, it is possible that using the NHI as a height threshold will possibly remove more data than intended.

### 3.3 Fourier transform wind retrieval

As described in subsection 2.1, the WindCube scanning strategy used during the campaign produced observations at five azimuthal angles that differ from the four azimuthal angles ($0°$, $90°$, $180°$, $270°$) used for the Doppler-Beam-Swing strategy (Röttger and Larsen, 1990). Similarly, CLARA produced continuous MDV observations in PPI mode.

In order to retrieve the wind speed and direction profiles from both sets of observations, one can use the velocity azimuth display (VAD) method (Doviak and Zrnic, 2006; Browning and Wexler, 1968). This approach assumes that horizontal wind is uniform within the scanned volume and that the vertical velocity of the scatterers is the same for all azimuths. Under those assumptions, the radial velocity can be described as a Fourier series, but only the first coefficient is used for determining the wind speed and direction (Doviak and Zrnic, 2006; Browning and Wexler, 1968).

In this work, the wind speed and direction profiles are derived using the Fast Fourier Wind Vector Algorithm (FFWVA) proposed by Ishwardat (2017), and a brief description of this method is given below. The FFWVA is similar to the VAD method. However, instead of using the Fourier series, it takes advantage of the currently available fast Fourier transform algorithms for digital signal processing to decompose the radial Doppler observations in terms of amplitude and phase of their harmonic frequencies. Note that the unit of these frequencies is 1/deg and not 1/time. The amplitude and phase from the first harmonic are used for calculating the wind speed and direction, and the determination of both quantities is summarised as follows:

$$a + bi = \text{DFT}(V(\phi))|_{1st} \tag{3}$$

$$\phi_d = -\arctan(\frac{b}{a}) + 180 \tag{4}$$

$$V_h = \frac{2|a+bi|}{N\cos(\alpha)} \qquad (5)$$

$a$ and $b$ are the real and the imaginary parts from the first harmonic, respectively. $V(\phi)$ is the azimuthal slanted MDV values from one complete scan, and $\phi$ is the azimuth. DFT stands for discrete Fourier transform. $\phi_d$ is the wind direction azimuth related to the North. $N$ is an amplitude correction parameter, and its value is equal to the number of data points used in the transformation. $\alpha$ is the scanning elevation angle. Using this method, the retrieved wind will lose information from spatial variability smaller than a complete scan. The energy of small-scale variability will be distributed among the higher harmonics. In principle, it would be possible to use the higher harmonics information to identify the regions and periods of enhanced horizontal wind variability. Since the sampled volume diameter increases with height, it is expected that the variability of the horizontal wind within the sampled volume will increase with height. Consequently, the distribution of energy towards higher harmonics will increase with height.

### 3.4 Radar observations below 2 km

In addition to the data obtained from the cloud layer, CLARA also received echoes from the sub-cloud layer during clear air conditions and precipitation. Due to CLARA's polarimetric capability, it was also possible to obtain the differential reflectivity ratio (ZDR) from the sub-cloud layer while executing the PPI scans at 75° elevation. To investigate the origin of those sub-cloud layer echos and whether the retrieved wind information was biased, the maximum ZDR from each PPI scan is combined with the difference between wind speed derived from the WindCube and CLARA's data from the entire campaign.

The distribution of maximum ZDR stratified by height shows two main regions (Figure 6). The first region is above 2 km, where the histogram shows one main mode at 0 dB and a broadening of the distribution up to 7 dB. In this region, the temperature is colder than 0 °C, suggesting that the ZDR signal is likely produced by pristine ice crystals, snow and super-cooled liquid water. In the second region, below 2 km, the distribution shows two main modes, the first at 0 dB and the second at 7 dB. In this region, the temperature is warmer than 0 °C suggesting that the mode at 0 dB is likely to be produced by water droplets. However, what could be producing the second ZDR mode at an elevation of 75°? Previous studies have already indicated that radar returns from clear air conditions are likely to be from insects (Chandra et al., 2010; Geerts and Miao, 2005; Ritvanen et al., 2022) and also that insects could produce a strong polarimetric signature in the boundary layer region (Wainwright et al., 2017; Achtemeier, 1991; Wilson et al., 1994; Rennie et al., 2010; Martner and Moran, 2001).

Results from previous studies suggest that insects may actively fly and not only be carried by the wind, and due to it, the derived wind information could be biased (Lhermitte, 1969; Achtemeier, 1991; Wainwright et al., 2017; Chandra et al., 2010). On the other hand, the study from Wilson et al. (1994) suggests that the wind carried the insects, and no systematic bias was found. Klingebiel et al. (2019) combined 35.5 GHz cloud radar and lidar observations from Barbados and identified that radar returns (from -65 to 50 dBz) from below non-precipitating clouds coincide with regions where the lidar depolarisation ratio indicate the presence of spherical-like scatterers. The authors suggest that the radar returns are likely from sea salt. Based on radar observations and without in-situ measurements, we cannot affirm that the clear air echoes from the lowest 2 km are

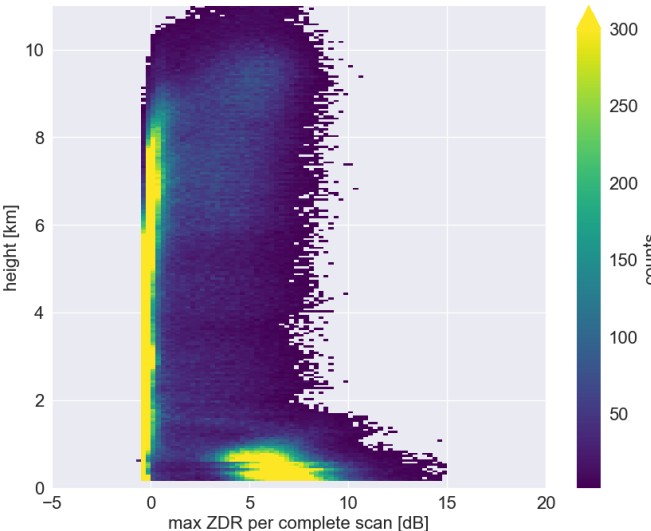

**Figure 6.** Two-dimensional histograms of the maximum ZDR for each PPI scan from the entire CMTRACE stratified with height. Note that only data with $I_{inv}$ smaller than 50% is used.

from insects or wet aerosols. Nevertheless, to evaluate if CLARA's retrieved wind speed is biased, the relationship between the retrieved wind speed difference (CLARA - WindCube) and the ZDR was investigated using data from the lowest 2 km

only. Figure 7 shows that the difference in velocity for ZDR larger than 4 dB is similar to that from ZDR around 0 dB, and no systematic bias is found. These results suggest that, over Cabauw, the clear air scatterers are likely to be carried by the horizontal wind, giving the possibility to use them to retrieve information from the horizontal wind. Nevertheless, if those scatterers are insects, they can fly up or downward actively (Rennie et al., 2010; Achtemeier, 1991; Chandra et al., 2010), but it was not investigated in this study.

Under precipitation conditions and in the presence of vertical wind shear, previous studies suggested that due to the differential fall velocity, smaller droplets can be advected further away than larger droplets (Laurencin et al., 2020; Dawson et al., 2015; Kumjian and Ryzhkov, 2012; Biswas et al., 2022), suggesting that larger droplets are less affected by the horizontal wind. Consequently, the horizontal wind retrieved from observations collected during precipitation may differ from the real wind. Under drizzle, the vertical wind component observed by the wind lidar may also differ from the real magnitude. Those

small droplets' fall velocity could contribute to the lidar observed vertical velocity. Additionally, Ghate et al. (2021) suggested that the evaporative cooling effect induced by drizzle strengthens the air's downward motion and weakens the upward motion.

Besides wind profiles, the dataset contains Ze and MDV profiles from the MARA. We suggest that the dataset users combine MARA's Ze and MDV variables to flag precipitating periods.

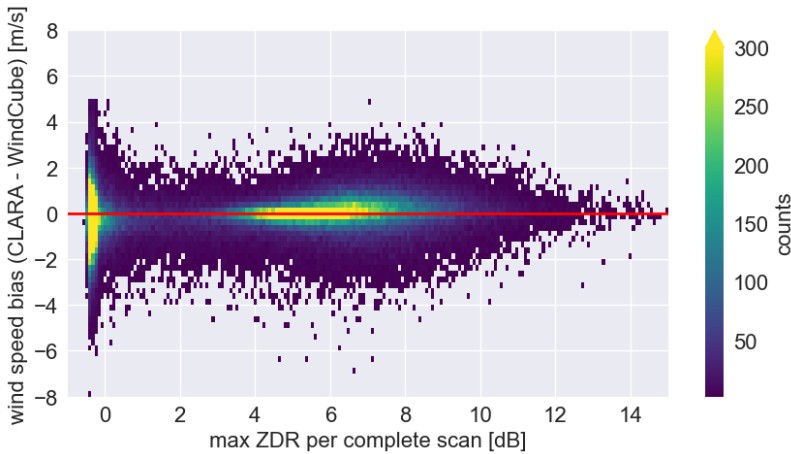

**Figure 7.** Two-dimensional histograms of the difference between WindCube's and CLARA's wind speed from the lowest 2 km as a function of the maximum ZDR. Note that only data with $I_{inv}$ smaller than 50% is used.

## 3.5 Radar wind direction offset estimation

After the campaign, it was noticed that the consecutive wind direction profiles from CLARA were periodically biased. In order to assess this bias, direction profiles from CLARA were compared with the direction profiles retrieved from the WindCube data. The histogram of the differences between both sets of profiles from the entire experimental campaign (Figure 8-(a)) shows that most points are distributed below 2 km, and the distribution is much broader in this region if compared with the region above 2 km. The probability distribution of differences (Figure 8-(b)) shows that the maxima are constant within each

range interval from each chirp sequence, suggesting that the biases are constant in those regions. It also shows that the biases are positive for some profiles, and for others, the biases are negative.

  The biases from each range region were calculated as the mean of all probability distribution maxima within each region. The retrieved values are $\pm 1.4°$ for the first chirp, $\pm 3.2°$ for the second, and $\pm 5.4°$ for the third chirp; the red dashed lines in Figure 8-(b) indicate those values. The negative biases are from profiles when the scans increase in azimuth from 0 to 359°,

and the positive biases are from scans when the azimuths decrease from 359 to 0°. Figure 12-(b) shows the result of the offset correction, and one can see that the bias along the range coordinate is close to 0°. The possible reason for this range dependent offset is that it takes around 1 s for CLARA to sample the 3 range intervals consecutively while CLARA rotates with an angular speed of approximately 5 $°\mathrm{s}^{-1}$. Then, the data from each range interval is stored, and the azimuth from when the sampling started is assigned to them, even though each range interval was sampled at a different azimuth.

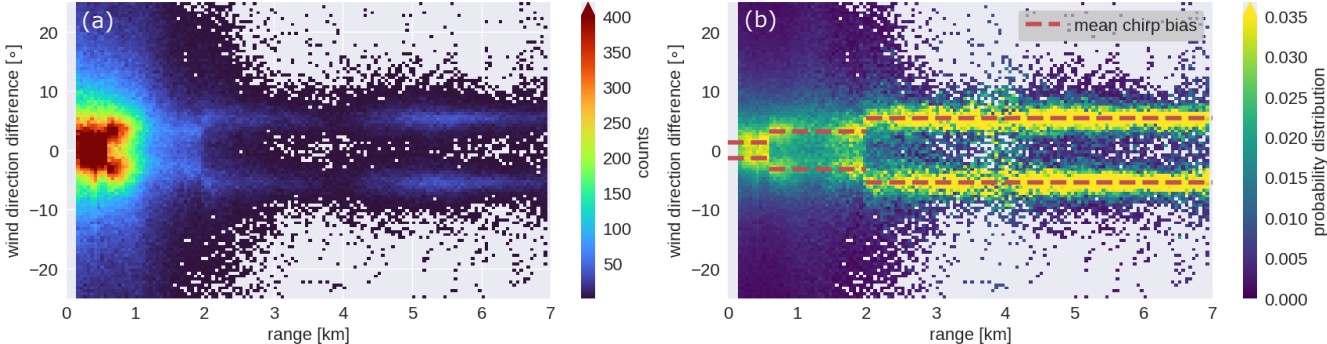

**Figure 8.** Panel (a) shows the two-dimensional histogram of the wind direction differences (CLARA - WindCube), and Panel (b) shows the probability distribution of the differences. The data used in both panels is from the entire CMTRACE campaign. The red dashed line indicates the mean bias for each range interval.

## 4 Data Evaluation

### 4.1 Radiosonde comparison to lidar and radar

In order to evaluate the wind speed and direction retrieved from the WindCube and CLARA observations, profiles of both observables from 34 radiosondes launched in De Bilt were used; the launching site in De Bilt is approximately 25 km away from the remote sensing site in Cabauw. For the evaluation, the Level 1 data from the WindCube and CLARA was used, but the High-STE filter was not applied to the WindCube observations in order to evaluate the agreement between the observations from regions above 4 km. For the evaluation, the WindCube and Clara profiles were averaged within a time window of 10 minutes centred at the launching time. Figure 9 shows an example of those profiles from September 27[th], 2021. Even though both sites are far from each other, the WindCube and CLARA profiles surprisingly almost overlap the radiosonde profiles.

A statistical analysis combining all radiosonde profiles suggests that the WindCube and CLARA observations of wind speed and direction are comparable to the radiosonde measurements (Figure 10); it is also supported by the statistical metrics bias, RMSE and correlation listed in Table 6. A more precise evaluation of the WindCube and CLARA data near the surface can be made using the measurements from the KNMI mast tower; however, the mast measurements are not available for this publication.

### 4.2 Radar and lidar inter-comparison

In addition to the data evaluation using radiosonde observations, the WindCube and CLARA profiles from the entire dataset were compared against each other. Analogous to the previous subsection 4.1, the High-STE filter was not applied to the WindCube data, in order to assess the quality of the wind profiles in the cloud layer.

The statistical analyses of wind speed (Figure 11-(a)) and direction (Figure 11-(b)) reveal that the observations from both instruments are well correlated with relative small bias, $0.24°$ for wind direction and $-0.16\,\mathrm{ms^{-1}}$ for wind speed. The agreement

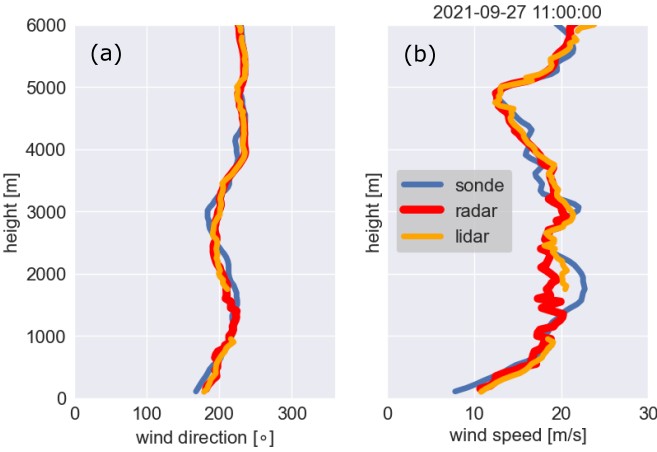

**Figure 9.** Profiles of wind direction (panel a) and wind speed (panel b) from 27.09.2021 at 11:00 UTC. The blue, red, and orange lines indicate the radiosonde, CLARA and WindCube observations, respectively.

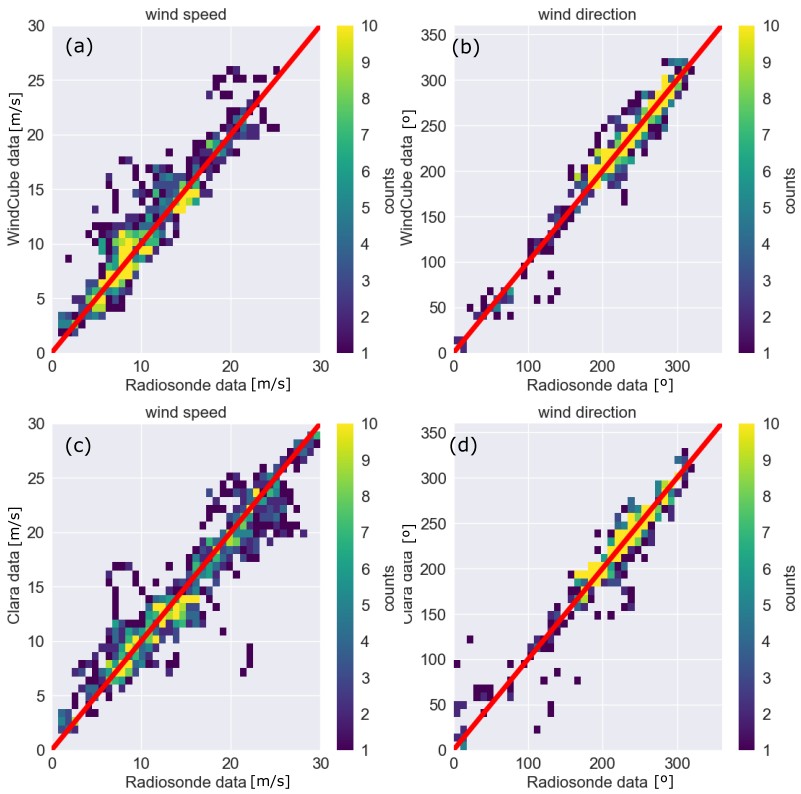

**Figure 10.** Two-dimensional histograms of wind speed (panels a and c) and direction (panels b and d) data retrieved by the WindCube (panels a and b) and CLARA (panels c and d) versus all available radiosonde measurements.

**Table 6.** Statistical metrics of the WindCube and CLARA data assuming the radiosonde measurements as a reference. RMSE stands for Root Mean Square Error. The bias and RMSE are in $\mathrm{ms}^{-1}$ and degrees for the windspeed and wind direction, respectively.

| metrics | WindCube | | Clara | |
|---|---|---|---|---|
| | wind direction | wind speed | wind direction | wind speed |
| bias | 0.37 | 0.52 | -0.24 | -0.34 |
| RMSE | 12.62 | 1.98 | 14.03 | 2.35 |
| correlation | 0.98 | 0.92 | 0.96 | 0.94 |

between the data from both instruments is also supported by the statistical metrics listed in Table 7. In addition, Figure 12-(a,b) show the difference between CLARA and WindCube observations of wind speed and direction stratified with range. Those figures show that most of the differences are distributed around 0 for both variables, indicating that the observations provided by both instruments are comparable. They also show that, at 3 km, both difference distributions broaden towards the surface, indicating that CLARA observations deviate from the WindCube observations. As suggested in subsection 3.4, insects are

likely the source of information in the lowest 2 km. It is possible that insects' random motion is increasing the uncertainty of the retrieved wind profiles.

Given the good agreement between the observations from both instruments, CLARA observations of wind speed and direction from regions below the NHI were included in the merged Level 2 dataset if the WindCube did not provide them. The impact of the broadening of the difference may not be significant for the Level 2 data because most of the observations in the

345 lowest 3 km are from the WindCube as shown in Figure 13.

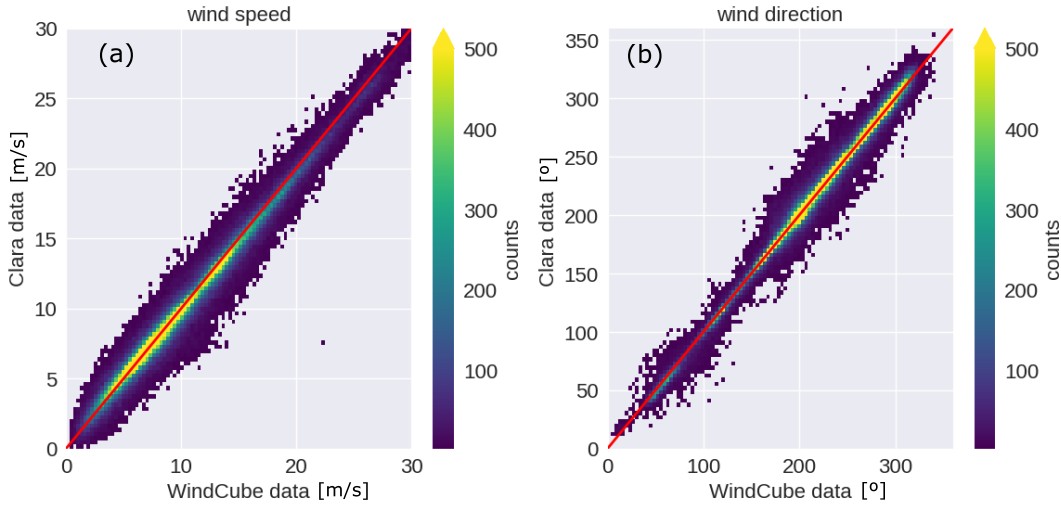

**Figure 11.** Two-dimensional histograms from the data pair WindCube-CLARA for wind speed (Panel a) and wind direction (Panel b); the red line indicates the 1 to 1 line.

**Table 7.** Statistical metrics of the intercomparison between CLARA and WindCube observations. RMSE stands for root mean square error. The bias and RMSE are in $\mathrm{ms}^{-1}$ and degrees for the windspeed and wind direction, respectively.

| | CLARA x WindCube | |
|---|---|---|
| metrics | wind direction | wind speed |
| bias | 0.24 | -0.16 |
| RMSE | 12.85 | 0.93 |
| correlation | 0.98 | 0.99 |

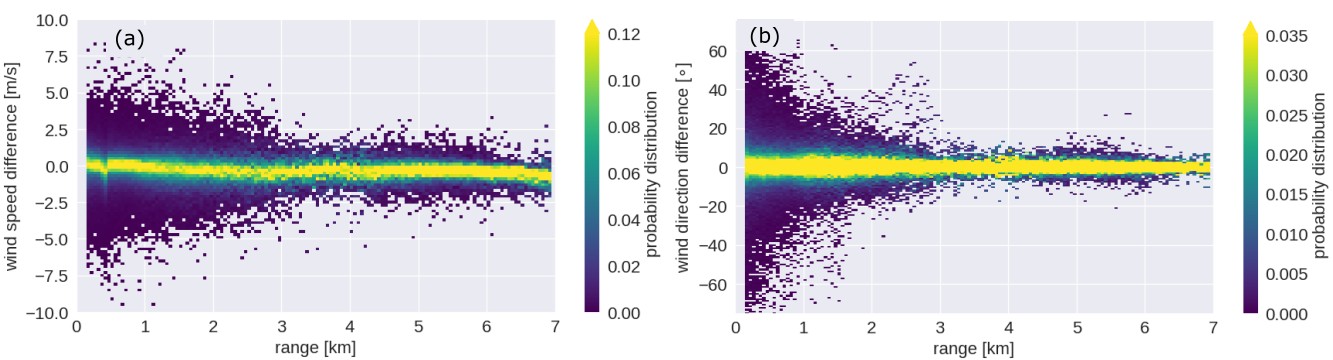

**Figure 12.** Two-dimensional probability distribution of the wind speed (Panel a) and direction (Panel b) differences stratified with height.

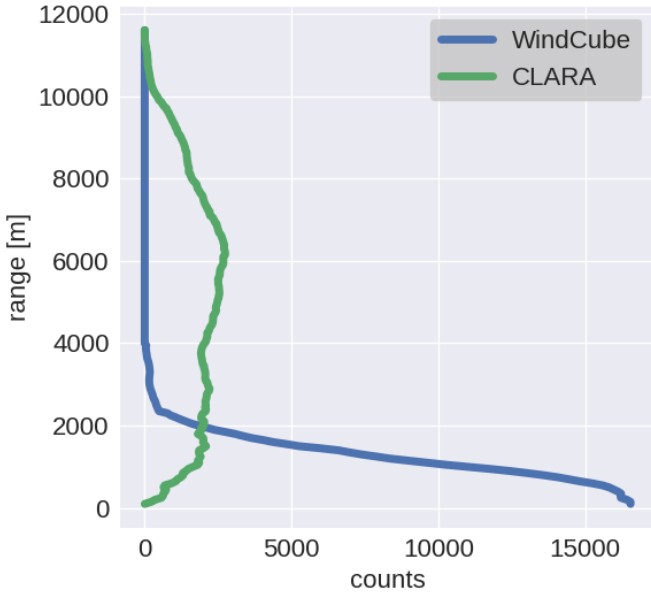

**Figure 13.** Distribution of Level 2 data source stratified with height: WindCube in blue and CLARA in green.

## 5   Application highlights

The CMTRACE dataset may be used for different applications. Here we would like to highlight two ways in which we are currently using the data: 1) the validation of wind and momentum transport in models; and 2) process studies of convective momentum transport (or transport of scalars through the boundary layer). A third application, which we do not further exemplify here, is the assimilation of wind profiles by weather models.

### 5.1   Model validation

The DALES model is running in near real-time hindcast mode on a (still) relatively small domain (15 x 15 km) centred at Cabauw with a grid spacing of 75 m. As part of the Ruisdael Observatory, these simulations will, in the future, be running on domains spanning the Netherlands at a grid spacing of ≈100 m. Along with model output at Cabauw from KNMI's mesoscale weather model HARMONIE, Figure 15 shows an initial comparison of observed and modelled winds on September 29[th], 2021. On this day, a frontal system passed over the observational site from the early to late morning. The Ze (Figure 14-(a)) and the MDV (Figure 14-(b)) recorded by MARA show that between 00:00 and 12:00 UTC, precipitating deep clouds with stratiform outflow passed over the site, while after the front passage, shallow cumulus and congestus clouds developed with tops between 4 - 7 km.

The Level 2 wind direction (Figure 15-(a)) and speed (Figure 15-(c)) show that before 06:00 UTC, the wind was mainly southerly throughout the lower and middle troposphere. During the frontal passage (06:00 - 12:00 UTC), the wind direction changed from southerly to westerly, while winds of 10 - 15 $\text{ms}^{-1}$ extended from near the surface up to cloud tops. After 12:00 UTC, the horizontal wind was mainly westerly, and the wind speed in the lower boundary layer was faster at about 15 - 20 $\text{ms}^{-1}$. The DALES model simulates the front passage and post-frontal convection (Figure 15-(b,d)), but some differences are apparent. The frontal passage seems to be slower in the model, judging mainly from the slanted wind direction and wind speed signature. Furthermore, in the period of post-frontal shallow convection (12:00 - 18:00 UTC), wind speeds in the observations appear to reach values up to 20 $\text{ms}^{-1}$ (orange-red) that can extend all the way to the surface while DALES maintains weaker surface layer winds. Comparing winds at 0.1 km above the surface (Figure 16-(a),(b)) reveals that the wind turning associated with the front passage in DALES is well simulated, but the wind speeds at 100 m are evidently too weak. Because the DALES winds showed are averages over the 15 x 15 km domain, while the observations represent wind variability at a single location, DALES is not expected to show the convective variations. However, in the observations, such convective variations would average out over time, and the weak wind bias in DALES appears too persistent throughout the day to be caused by the difference in domain-averaged versus point estimates.

The bias is intriguing because earlier comparisons of DALES to observations revealed a too strong wind bias instead (van Stratum et al., 2019). DALES uses a surface roughness length that underestimates the effective regional roughness length at Cabauw, which leads to too small surface stress. Ongoing investigations also include HARMONIE and employ measurement simulators to ensure a fair comparison of observations to models.

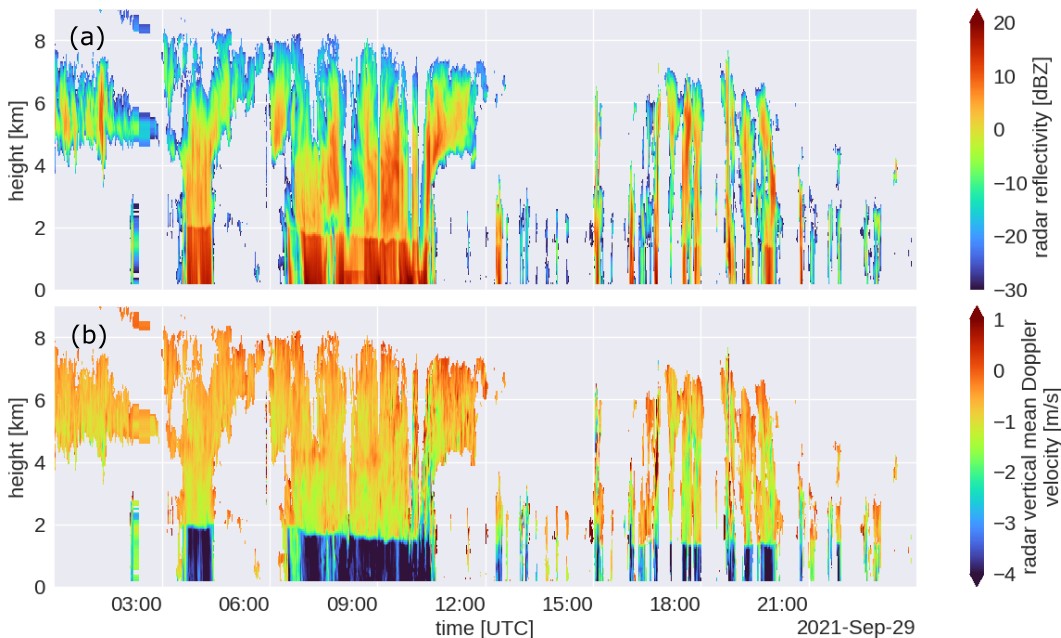

**Figure 14.** Time-height plot of MARA's observations from September 29th, 2021: (a) attenuated equivalent radar reflectivity, (b) mean Doppler velocity (negative means towards the ground).

### 5.2 Momentum transport on different scales

Focusing on the period of post-frontal shallow convection on the same day (Figure 17), we can illustrate the presence of multi-scale flows in the convective boundary layer. Reynolds averaging (Stull, 2003) is applied to the vertical and horizontal wind using a sliding window to separate flows with scales longer and shorter than 10 minutes (Equation 6). $v_\mathrm{obs}$ is the observed wind, $\bar{v}_\mathrm{10m}$ is the 10 minutes averaged wind, and $v'_\mathrm{10m}$ is the 10 minutes anomaly.

$$v'_\mathrm{10m} = v_\mathrm{obs} - \bar{v}_\mathrm{10m} \tag{6}$$

Assuming wind speeds of 10-20 $\mathrm{ms}^{-1}$, a 10-minute average window corresponds to a spatial scale of about 6-12 km, which is in the meso-$\gamma$ range. Figure 17 a) and b) illustrate the fluctuations in the horizontal and vertical wind speed on these scales, while Figure 17 c) and d) show the presence of convective up- and downdrafts that carry different winds. Evidently, during the frontal passage, before 12:00 UTC, the vertical motion is on average downward (red) in the presence of precipitating convection, even though individual updrafts and downdrafts are visible. MARA's observations from the same period and height show an intensification of Ze and MDV (Figure 14-a, b), which suggests the occurrence of precipitation. It is possible that $\bar{w}_\mathrm{10m}$ not only contains information from downward winds, but also from the terminal fall velocity of raindrops (Aoki et al., 2016).

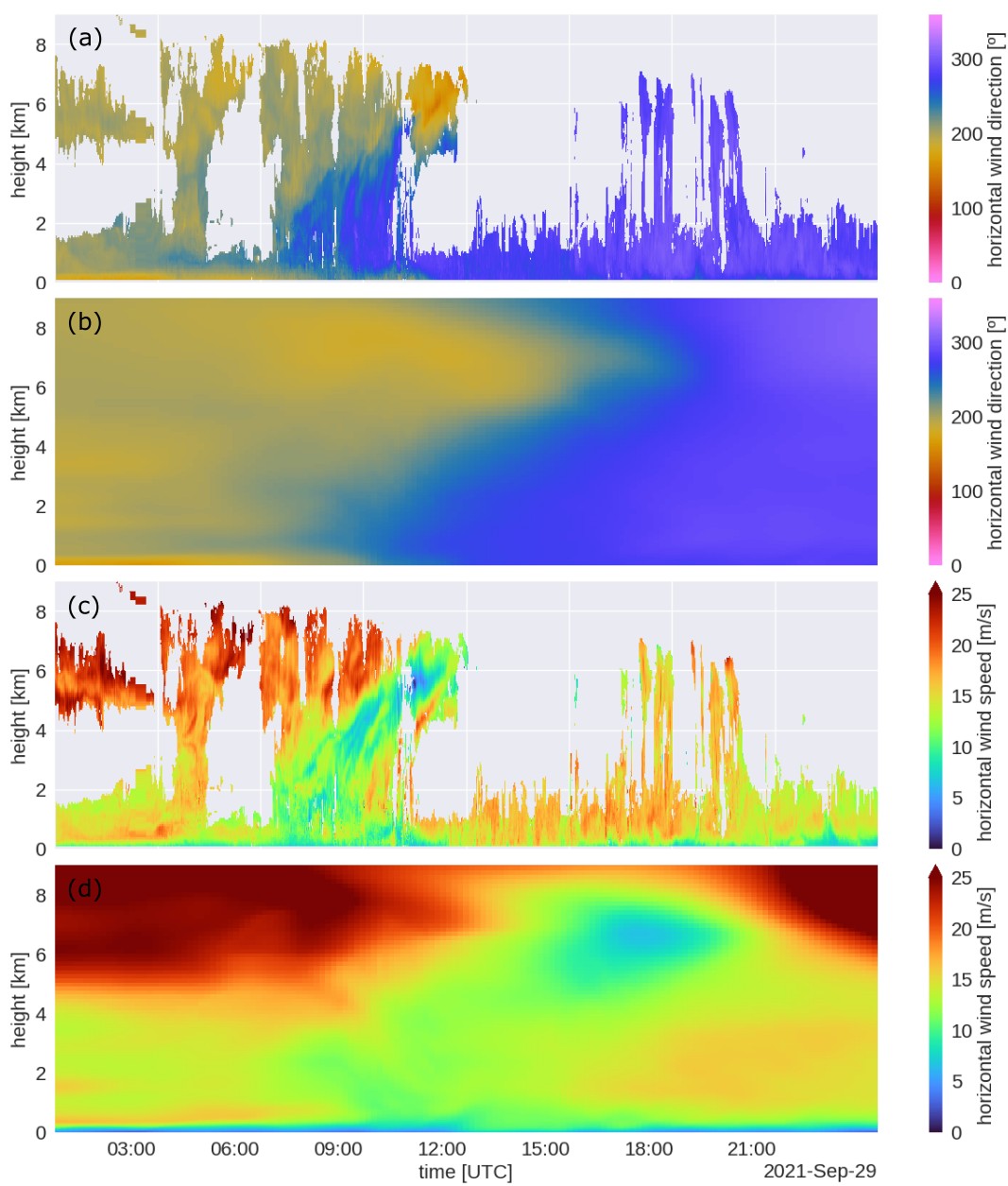

**Figure 15.** Time-height plot of wind direction (panels: a and b) and speed (panels: c and d) from the CMTRACE Level 2 (panels: a and c) and from DALES (panels: b and d) from September 29th, 2021.

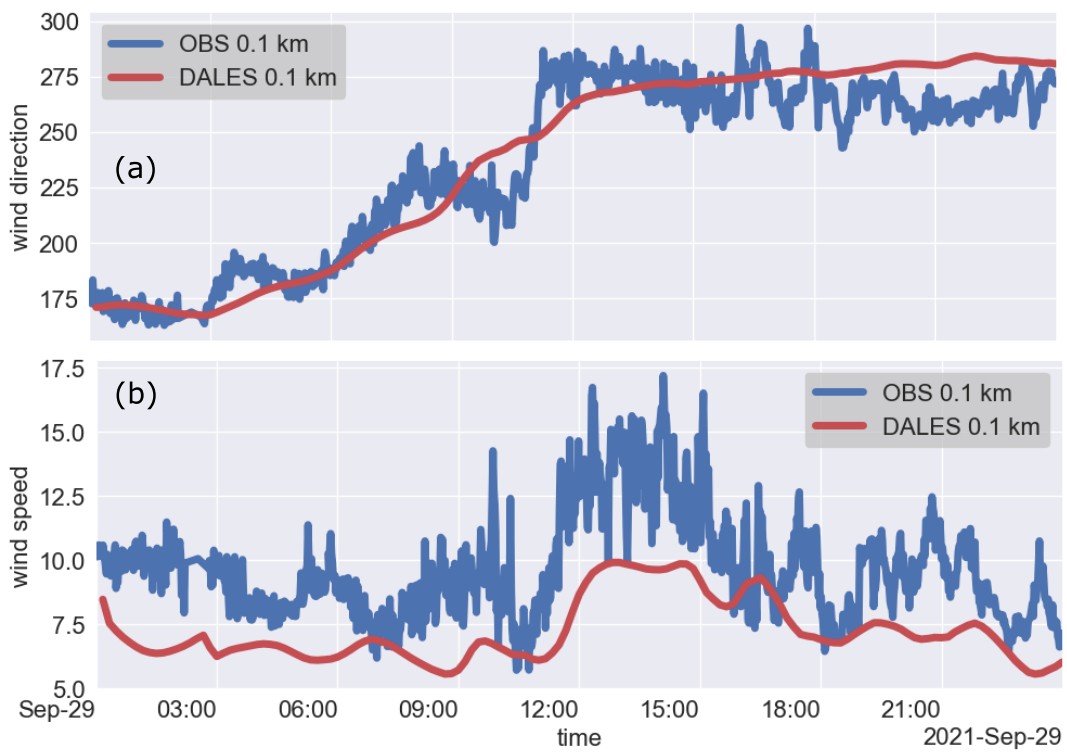

**Figure 16.** Time series of wind direction (panel a) and speed (panel b) from September 29th, 2021 at 0.1 km above the surface. The blue curve is from the lowest available observation in the CMTRACE Level 2; the red is from DALES.

After 12 UTC, mesoscale fluctuations in both the vertical and horizontal wind are evident, where downward motion tends to be accompanied by stronger horizontal winds extending from the top of the boundary layer to the surface and upward motion to weaker horizontal winds extending upward from the surface. Qualitatively comparing the horizontal and vertical speed observation with MARA's Ze suggests that faster horizontal winds and downward motion are associated with periods of congestus and precipitation. Drizzle, evaporative cooling and associated downdrafts, as suggested by Ghate et al. (2021), may contribute significantly to the momentum transport. In turn, slower horizontal winds and upward motion are associated with periods of no precipitation and clear skies. These mesoscale-like fluctuations may be coupled to an organization in the cloud field - a currently popular subject of research in the cloud - circulation - climate community - and contribute non-negligibly to total momentum transport. Past LES modelling studies, including with DALES, have not been able to accurately represent these mesoscale flows due to their use of small domain sizes and periodic boundary conditions. As DALES will be running on much larger domains with open boundary conditions, the ability to validate such flows with observations is necessary. In our current ongoing work, we use spectral analysis to derive the contribution of convective and mesoscale fluctuations to total momentum transport.

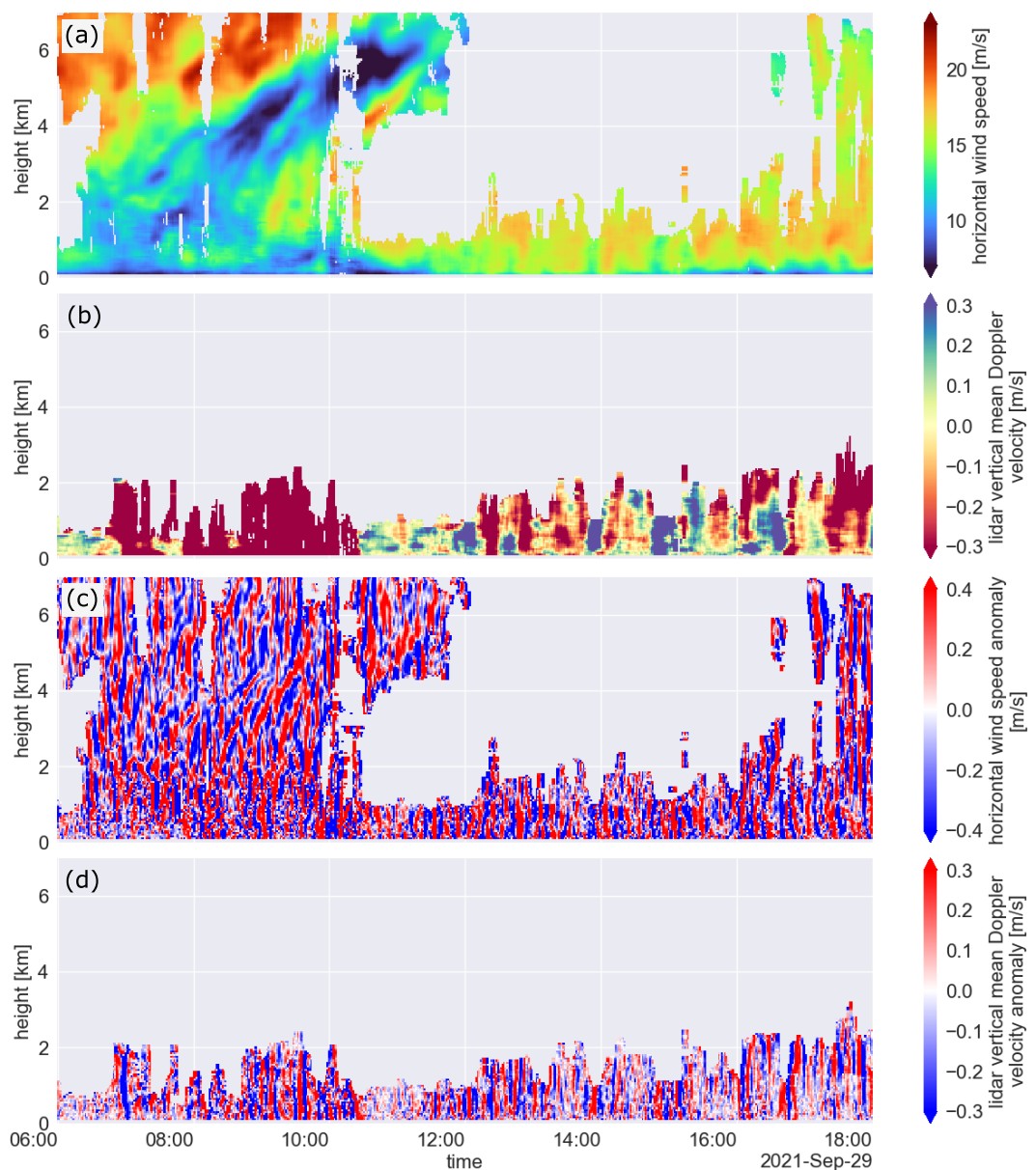

**Figure 17.** Time-height plots from September 29th, 2021: (a) CMTRACE Level 2 horizontal wind speed main component (10 minutes averaged), (b) CMTRACE Level 2 lidar vertical mean Doppler velocity (10 minutes averaged, negative means towards the surface), (c) CMTRACE Level 2 horizontal wind speed anomaly, (d) CMTRACE Level 2 lidar vertical mean Doppler velocity anomaly.

## 6 Conclusions

This manuscript introduces an experimental setup for retrieving horizontal wind speed and direction profiles from near-surface to cloud top, taking advantage of the synergy of using wind lidars to retrieve wind profiles in the boundary layer and cloud radars to retrieve wind in the cloud layer. The first CMTRACE campaign was conducted in Cabauw, the Netherlands and lasted 21 days, and its results are presented here. During this experiment, most days were non-precipitating with the presence of few clouds (mainly low level clouds). The winds at 200 m were mainly southwesterly, with speeds between 3.6 and 12.9 $\mathrm{ms}^{-1}$.

The CMTRACE dataset was processed in two levels. In Level 1, the processing minimizes the presence of second trip echoes that affect the WindCube measurements, and reduces the bias in wind direction profiles derived from the radar observations. The Level 1 observations from each instrument are kept in their original spatial and temporal resolution. In Level 2, the lidar and radar differences in the sampled volume are reduced, and the horizontal wind profiles from both instruments are merged, creating a single profile with a flag to identify the measuring instrument.

Statistical analyses are used to assess the level of confidence of the CMTRACE dataset. When correlating the CMTRACE data with radiosonde measurements, the results show that this correlation is higher than 0.9 for wind speed and wind direction. The results also indicate that the absolute wind speed bias is smaller than 0.55 $\mathrm{ms}^{-1}$, and the absolute wind direction bias is smaller than 0.4° (Table 6). The surprising possibility of using CLARA for retrieving wind profiles in the boundary layer, likely due to the presence of insects, allows the agreement between lidar and radar measurements to be evaluated. The intercomparison 420 between lidar and radar reveals that for wind speed and direction, the correlation is higher than 0.95 (Table 7), and the bias of wind direction and speed is 0.24° and -0.16 $\mathrm{ms}^{-1}$, respectively.

Possible applications of the CMTRACE dataset include model validation (*e.g.*, convection-permitting model simulations), analysis of the scales of motion that accompany diverse (organized) cloud fields, process studies of momentum transport, and transport of scalars (*e.g.*, air pollution). The subsequent CMTRACE campaigns are intended to last several months in a row to 425 capture a diverse cloud regime and periods when congestus clouds are predominant. An extension of the experiment to other regions (*e.g.* in the tropics) is also planned.

## 7 Data availability

The CMTRACE Level 1 and Level 2 datasets are publicly available on the ZENODO platform. CMTRACE Level 1 can be downloaded from https://doi.org/10.5281/zenodo.6926483 (Dias Neto, 2022a) and Level 2 from https://doi.org/10.5281/zenodo.6926605 430 (Dias Neto, 2022b). The Level 0 can be requested from the corresponding author.

## Appendix A: List of variables

The files from the Level 1 and Level 2 data are structured differently. As described in section 3, Level 1 data from each instrument is kept with its original spatial and temporal resolution. The Level 1 data from the instrument was resampled to a common spatial and temporal resolution and used to generate the Level 2 data. The data from both levels are stored as NetCDF files. The variables available in Level 1 and Level 2 data are listed in Table A1.

*Author contributions.* JDN wrote the paper, developed the data processing, and designed the experiment. LJ supervised, designed the experiment and reviewed the manuscript. CU managed the radars, designed the experiment, participated to the supervision and reviewed the manuscript. SK managed the lidar, designed the experiment and reviewed the manuscript.

*Competing interests.* The authors declare that they have no conflict of interest.

*Disclaimer.* TEXT

*Acknowledgements.* This publication is part of the NWO Talent Scheme Vidi project CMTRACE with project number 192.050, financed by the Dutch Research Council (NWO). We acknowledge the Ruisdael Observatory for providing the infrastructure for this experiment. We acknowledge the Royal Netherlands Meteorological Institute for providing the radiosonde and ceilometer observations (https://dataplatform.knmi.nl/) and support at the Cabauw site. We thank Saverio Guzzo from Delft University of Technology for managing the network and the data transfer. We also thank Robert Mackenzie from Delft University of Technology for mounting the instruments on the experimental site and operating it during the experiment. Finally, we thank two anonymous reviewers for their comments on the first draft.

**Table A1.**

| Variable Name | Level 1 | | | Level 2 | description |
|---|---|---|---|---|---|
| | WindCube | Clara | Mara | merged dataset | |
| horizontal_wind_direction [°] | x | x | | x | horizontal wind direction retrived using the FFT method with respect to true north |
| horizontal_wind_speed [m/s] | x | x | | x | horizontal wind speed retrived using the FFT method |
| zonal_wind [m/s] | x | x | | x | zonal wind retrived using the FFT method |
| meridional_wind [m/s] | x | x | | x | meridional wind retrived using the FFT method |
| vertical_wind_speed [m/s] | x | | | x | observed vertical wind speed (negative towards the ground) |
| lidar_relative_beta [m$^{-1}$sr$^{-1}$] | x | | | x | Attenuated relative backscatter coefficient from the vertical beam |
| start_scan [s] | | x | | | starting time of each scan used for generating each profile |
| end_scan [s] | | x | | | end time of each scan used for generating each profile |
| nan_percentual | | x | | | percentual of invalid data per complete PPI scan |
| max_radar_differential_reflectivity [dB] | | x | | | maximum ZDR from each complete PPI scan |
| radar_vertical_doppler_velocity [m/s] | | | x | x | Doppler velocity observed by a vertically pointing 94 GHz radar (negative towards the ground) |
| radar_equivalent_reflectivity [mm$^6$/m$^3$] | | | x | x | attenuated radar linear equivalent reflectivity factor observed by a vertically pointing 94 GHz radar |
| radar_spectrum_width [m/s] | | | x | | Measure of the dispersion of radial velocity within the radar measurement volume |
| radar_spectrum_skewness | | | x | | Skewness of the velocity distribution within the radar measurement volume |
| rain_rate [mm/h] | | | x | x | rain fall rate measured by a near by PARSIVEL (optical disdrometer) |
| data_flag | | | | x | instrument identification flag: 0: no data, 1: WindCube lidar, 2: CLARA 35 GHz radar |

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
