# Peer review of "Combined Wind Lidar and Cloud Radar for High-Resolution Wind Profiling"

_Earth System Science Data, 2022_

## Referee Comment (RC2)

Review for: ***"Combined Wind Lidar and Cloud Radar for Wind Profiling"*** (essd-2022-268) submitted to *Earth System Science Data* by *José Dias Neto et al.*

**General recommendation:**

Minor revision

**Synopsis:**

The manuscript presents a dataset of wind profiles derived from scanning ground-based Doppler lidar and Doppler cloud radar observations. After presenting the measurement setup, the data are evaluated against each other and with radiosondes. In the end, some applications for this dataset are presented. The manuscript is well written and all necessary information to the dataset is provided. I have only some minor comments before I recommend this work to be published.

**General comments:**

- Section 2.1: For clarification, I would like to know if both the CLARA radar and the Doppler lidar were continuously performing these scans that were used for this study or if there were other observations (with other azimuth/elevation angles) in between. Please mention that shortly in this section!

- Like tables 1-3 it would be nice to have a table with the resolutions and range of the combined dataset
- To avoid second-trip echoes, it would it be possible to use a lower pulse repetition frequency. Is that possible with you lidar? If yes, would that have other negative implications? Please comment on that in the paper!

**Detailed comments:**

- Fig. 10, 11: please provide units for wind speed.
- p. 20, l. 325: You write about a comparison of observed and modelled winds in Figure 14, however I can only find observations. Please check!
- p.21, l. 355-359: Did you try to discriminate rain from wind? I guess, for the radar this is not possible. Therefore, I also think that the information in Fig. 17 c and d during the rainy period is not very useful. Please comment on that!
- The time variable in the dataset is given in nanoseconds after the first data point. I would strongly suggest a more user-friendly time unit.

---

## Author Comment (AC1)

**General comments by the authors to the Reviewers**

**We thank the reviewers for the time and effort in reading the manuscript and for the constructive comments and suggestions, which we in the following address point by point (our answers are marked with blue font colour).**

**Reviewer #1**

The authors have used data collected by two scanning radars, and one scanning lidar to derive profiles of horizontal wind speed and direction. The derived winds are compared to those from the radiosondes, and from the lidar only for validation. The authors have further applied the technique to a case of frontal passage and evaluated the model's performance in simulating the winds. The article is detailed and well-written, and the developed technique can be used for deriving winds at other locations as well. Below I am mentioning few things that can further improve the manuscript.

 **Major Concerns:**

1) It is unclear to me why the authors have not used radar wind profilers (RWP). The RWP have been used for multiple decades now to derive winds and their backscatter is not that affected by hydrometeors unless it is heavy rain. The modern wind profilers also can sample much lower in the boundary layer (~50 m). Same is also true for SODAR. It will be much easier for your group to use RWP and SODAR rather than derive winds this manner. So I'd like to know the reasoning behind this, and at least some discussion in the article. Just to be clear, this has no bearing on the proposed technique and the rationale behind the article.

R: We agree with the reviewer that the reason for using this combination of instruments was not mentioned. We will clarify why those instruments were used in the reviewed manuscript's introduction. We want to retrieve wind profiles with the highest possible temporal and vertical resolution. First, to study the processes and second, to evaluate them in high-resolution models e.g. LES, which has a spatial grid of 50 - 100 m, so already much higher than what we can measure. The most important reason for combining a wind lidar and a cloud radar for wind profiling is that we want to capture processes with temporal scales smaller than 10 min: turbulent flows driven by convection. As indicated in the manuscript, the combined RADAR-LIDAR temporal resolution is ~1.2 min, and the LIDAR observations alone have a temporal resolution of 2 sec, allowing exploring even shorter time scales.

We agree that RWP and SODAR have a long history of profiling winds and providing insights into the boundary layer structure and its processes above the boundary layer. The specifications of the current commercially available RWP indicate that the range resolution (40-120 m) and height of the first range gate (70 - 100 m) are similar to that of our combined RADAR-LIDAR wind profiles. Similarly, the specifications of commercially available SODARs indicate that the range resolution is ~ 10 - 50 m, higher than RWP; the height of the first range gate is ~ 15 - 40 m, lower than what we currently have. Even though the range specifications from both instruments are similar or better, both RWP and SODAR need a long averaging time of ~10 min or longer. Additionally, SODARs are strongly affected by environmental acoustic noise, which rapidly decreases the signal-to-noise ratio with height, limiting the height of the maximum range gate and rapid attenuation of the acoustic waves. In ideal conditions, some SODARs can provide observations up to 1 km. We believe that it would be beneficial to have an RWP in the experimental site as a complementary instrument in the future; the RWP could provide wind observations in the absence of clouds or even aerosols.

2) The winds derived from the PPI sequence (radar or lidar) have some inherent limitation on the uncertainty of the retrieved winds. As the derived winds assume that they do not change over the domain where the observations are collected. So in your FFWVA algorithm, the energy will go into other harmonics rather than the first. Can you please elaborate on this.

R: We agree with the reviewer that deriving winds based on PPI observations introduce limitations to retrieved wind profiles. The basic assumption of this approach is that the vertical velocity and the distribution of scatters are the same for all azimuths, which may not always be the case. For example, suppose some of the azimuthal observations are from a region where downward motion is taking place, and the other portion of the observations are from regions where upward motion is happening. In that case, the sine curve's amplitude may not be the same for all azimuths, and then the retrieved wind velocity will deviate from the actual wind velocity. Inhomogeneities in the horizontal wind can also lead to the ideal sine curve deviation.

The energy of the mean wind component will be captured by the first harmonic. However, part of the energy due to inhomogeneities will be distributed with other harmonics rather than the first. In principle, it would be possible to use this spectral decomposition to identify periods/regions of enhanced inhomogeneities among the azimuthal observations. We have not done it yet, but it is planned for future study.

3) There are too many references that have probed horizontal winds from the weather radars, so cannot mention one. But please look at publications from Chandra, Rizkov, kumjian etc.

**R: We agree that weather radars have long been used to probe horizontal winds; this fact will be acknowledged in the introduction.**

4) The horizontal domain over which the winds are derived, the vertical resolution, and temporal resolution are all very critical. It will be great if you can tell us the impact of very low winds, and very high winds on your retrieval technique.

**R: We agree with the reviewer that those parameters mentioned are critical for the wind profile estimation. This kind of sensitivity analysis would need a direct comparison using in-situ observations. The observations from the nearby meteorological tower are not available yet, and during this campaign, no aircraft measurements were conducted either. For those reasons, we cannot yet indicate the impact of low and high winds.**

5) Lastly, Figure 14 shows rain echoes in excess of 20 dBz, so maybe you are looking at drops more than 1-2 mm in diameter. When viewed by a tilted radar axis these drops contribute some to the horizontal winds as they are carried by them due to shear. This also needs to be mentioned/explored. Uncertainties in these retrievals will finally determine how far off your model is and can potentially lead to inaccurate conclusions. Thanks.

**R: As indicated by the reviewer, the retrieved horizontal wind under precipitation conditions may not reflect the true wind. Due to the terminal fall velocity being related to the droplet sizes, small rain droplets will have a longer residency time when compared to larger droplets. Previous studies indicated that in the case of upward motion, the hydrometeors could be affected by size sorting and further increase the residency time of the small droplets (Laurencin et al., 2020; Kumjian & Ryzhkov, 2012). They also found that in the case of vertical wind shear, smaller droplets are advected further away than larger droplets indicating that those larger droplets are less affected by the wind. The horizontal wind retrieved during precipitation, available in the dataset, should be used with caution. We will discuss it in the manuscript to make readers aware. In addition to the wind profiles, the radar equivalent reflectivity and the mean Doppler velocity from the vertically pointing cloud radar are also available in the level-1 and level-2 data. Those variables can be combined and used to flag the precipitating periods. We also agree with the reviewer that this dataset can also be used to investigate further the effect of vertical wind shear and its effect on precipitation.**

6) Can you comment on how you discern Doppler lidar echoes that are from the insects, aerosols, and from the rain? This is a very important issue as it affects the wind determination. You are already mentioning the Wainwright paper for insects, I know of the Ghate et al. 2021 JAMC paper for the hydrometeors. This will affect results shown in Figure 17.

**R: At the current stage, we cannot decouple the contribution of the different lidar scatterers from the retrieved horizontal wind. However, in both data levels, the mean Doppler velocity and the radar equivalent reflectivity from the vertically pointing cloud radar are included; if combined, those variables can be used to flag periods/regions where hydrometeors are likely to disturb the wind retrieved from the lidar observations. The work from Ghate et al. 2021 indicates that the presence of drizzles affects the vertical motion due to their evaporative cooling effect. The authors indicate that downdrafts tend to strengthen and updrafts tend to weaken under drizzling conditions. The same may also be happening in this case. We will review the interpretation of figure 17 in the manuscript to consider the possibility of the evaporative cooling effect.**

**The laser beam from the WindCube lidar is extremely narrow compared to the width of the cloud radars beam. It suggests that for the same range resolution, the lidar sampled volume is smaller than the radar sampled volume. Based on the sampled volume differences, we can expect that the probability of finding insects within the lidar sampled volume is lower than within the radar sampled volume. It is possible that insects can also be contributing to the lidar backscattered signal, but taking into account the lidar PRF and integration time, it is likely that the insect contributions will be averaged out.**

7) I still find it very hard to believe that insects are moving with horizontal wind as insects have been shown to stay near the water/vegetation. Do these insects just get blown away by the winds over the day, and hence things clear out? This way you'd need a lot of insects to be generated each day! It can be imagined that insects have very small impact on vertical wind, but not horizontal winds. See Chandra et al. 2010 JAS for radar retrievals from insect echoes.

**We agree that we cannot know for sure. However, in the absence of hydrometeors, in clear air conditions, the radar returns are likely to be from insects, as suggested by previous studies (Chandra et al. 2010, Geerts and Miao 2005 -JAO TECH, Wilson et al. 1994 -JAO TECH). Those studies also indicate that insects introduce bias to the vertical motion recorded by vertically pointing cloud radars. In the experiment conducted in the marine environment (Barbados, Klingebiel et al. 2019 -JAS), the authors investigated the LDR recorded by a**

**ceilometer. They found small LDR values, indicating that those scatterers have a small aspect ratio. The authors suggested that the low LDR values are likely from sea salt. In our study, we used the ZDR to have an indication of the aspect ratio of the radar clear air scatterers. The ZDR was recorded at a high elevation angle (75 º), while the same radar recorded the mean Doppler velocity for retrieving wind profiles. We found that most of the radar ZDR values from below 2 km are higher than 4 dB, suggesting that they have a high aspect ratio (the major axis divided by the minor axis). In addition, the difference between the wind speed retrieved by radar and lidar does not indicate an offset. We agree with the reviewer that this result is intriguing, suggesting that those scatterers are blown away continuously. Based on the remote sensing observations and without in-situ observation, we cannot say for sure that those scatterers are insects. However, the high ZDR values from clear sky echos agree with what one would expect from insects. Another possibility could be wet aerosol with a high aspect ratio. We will be a bit more careful in the manuscript in describing these results and also allude to other options.**

**Minor Concerns:**

Figure 4: there is no red curve.

**R: The label was corrected**

Line 240, define DFT.

**R: DFT was defined**

Figure 11: Mention units.

**R:  The figures were corrected**

Line 312: What is NHI?

**R: NHI stands for noise height interface and  is defined in section 3.2**

**Reviewer #2**

The manuscript presents a dataset of wind profiles derived from scanning ground-based Doppler lidar and Doppler cloud radar observations. After presenting the measurement setup, the data are evaluated against each other and with radiosondes. In the end, some applications for this dataset are presented. The manuscript is well written and all necessary information to the dataset is provided. I have only some minor comments before I recommend this work to be published.

**General comments:**

**1)** Section 2.1: For clarification, I would like to know if both the CLARA radar and the Doppler lidar were continuously performing these scans that were used for this study or if there were other observations (with other azimuth/elevation angles) in between. Please mention that shortly in this section!

**R: Both instruments, the cloud radar and Doppler wind lidar, were operated following the scanning strategy described in the manuscript and no other scanning strategy was used. It will be explicitly indicated in the text.**

2) Like tables 1-3 it would be nice to have a table with the resolutions and range of the combined dataset

**R: Thank you for this suggestion! A table listing that information will be added to section 3.1, where the processing levels are described.**

3) To avoid second-trip echoes, it would it be possible to use a lower pulse repetition frequency. Is that possible with you lidar? If yes, would that have other negative implications? Please comment on that in the paper!

**R: We agree that reducing the pulse repetition frequency would reduce the occurrence of second-trip echoes, and we very much would like to do so. However, for our current version of the WindCube 200S, the pulse repetition frequency and the range resolution cannot be set independently (50m at 15 kHz, 75m and 100m at 10 kHz). We have been in touch with the company regarding this aspect and learned that it will be improved in future versions of this instrument. During the experiment, the range resolution was set to 50 m, fixing the PRF to 15 kHz. The 50 m range resolution was chosen to allow retrieving wind profiles with high vertical resolution and comparable to the cloud radar vertical resolution.**

**Detailed comments:**

1) Fig. 10, 11: please provide units for wind speed.

**R:  Both figures were corrected**

2) p. 20, l. 325: You write about a comparison of observed and modelled winds in Figure 14, however, I can only find observations. Please check!

**R: The right figure was referenced**

3) p.21, l. 355-359: Did you try to discriminate rain from wind? I guess, for the radar this is not possible. Therefore, I also think that the information in Fig. 17 c and d during the rainy period is not very useful. Please comment on that!

**R: This comment is similar to a comment from review 1. Please refer to the answer to comment 6 from reviewer 1.**

4) The time variable in the dataset is given in nanoseconds after the first data point. I would strongly suggest a more user-friendly time unit.

**R: Thank you for pointing this out! The data processing was developed using python and based on the xarray library. By default, xarray uses time in nanoseconds. We will search for an alternative approach to save the time variable in seconds and release an updated dataset version.**